# Seasonal Variation and Quality Assessment of the Major and Trace Elements of Atmospheric Dust in a Typical Karst City, Southwest China

**DOI:** 10.3390/ijerph16030325

**Published:** 2019-01-24

**Authors:** Yang Tang, Guilin Han

**Affiliations:** 1Institute of Geochemistry, Chinese Academy of Sciences, Guiyang 550002, China; tangyang@mail.gyig.ac.cn; 2School of Scientific Research, China University of Geosciences (Beijing), Beijing 100083, China

**Keywords:** atmospheric dust, trace elements, pollution, Guiyang, enrichment factor

## Abstract

Atmospheric dust plays an important role in bio-geochemical cycling and the ecological environment. In urban areas, atmospheric dust is more likely to be the carrier of pollutants, thus affecting the air quality of cities. In this study, samples of atmospheric dust were collected monthly for one year in Guiyang City, and the contents of major and trace elements in the dust were determined. The results showed that the major and trace elements in the atmospheric dust of Guiyang city vary with the seasons. The concentrations of trace elements in the dust of autumn and winter were significantly higher than those in spring and summer. Most of the major elements in dust were higher than those in the corresponding topsoil, while most trace elements were similar to those in the topsoil except for Pb. The enrichment factor (EF) values calculated by topsoil as background showed that the Ca and Pb have higher EF values than the rest elements, indicating that some dust may be contaminated by Ca and Pb. The high content of Ca in the dust might be derived from concrete buildings in urban areas, while Pb might be closely related to motor vehicle emissions. The relationship between La and Yb of the atmospheric dust showed that the dust in Guiyang have the same pattern as those of Libo, it also revealed that these dust probably come mainly from natural sources and are less affected by human activities.

## 1. Introduction

Atmospheric dust is a kind of raw material existing widely in the surface ecosystem. During geological history, atmospheric dust has been very important to the global bio-geochemical cycle [1,2]. Atmospheric dust is a medium for various elements, such as the nutrients carried by the Saharan atmospheric dust, which can promote the growth of algae in the Atlantic Ocean [3,4,5,6,7]. Atmospheric dust can also accumulate over a long period to reconstitute the surface topography of the region, and might even have constructed the Loess Plateau of Western China [8,9]. Atmospheric dust can also influence the local and global climate by affecting the incidence of the solar light during the transmission process [10].

The raising, transportation and deposition of dust are mainly of natural course, in which some anthropogenic materials are discharged into the air and mixed with natural dust [1]. In the past two hundred years, the process of industrialization and urbanization has been accelerating. More and more substances related to human activities are released into the atmosphere and become part of the atmospheric dust [11,12]. In urban areas, atmospheric dust is easily polluted by man-made substances, and most of these pollutants are heavy metals emitted by a series of human activities [13,14,15,16]. Some studies have shown that particulate matter in urban soils and roads is significantly affected by heavy metals produced by human activities [17,18,19,20], and these heavy metals may originate from the deposition of polluted atmospheric [21,22,23].

Numerous studies have proved that the air of many large cities in China is seriously polluted by heavy metals [24,25,26,27]. Dust itself and its contaminating elements may lead to a variety of human diseases for excessive exposing to or inhaling of atmospheric dust [28,29]. In this study, Guiyang City, which has relatively good air quality, was selected as the study area, the impact of natural and human activities on the atmosphere of Guiyang City is revealed by using dust heavy metal content index, The source and migration process of pollutants were also discussed, so as to distinguish the various influencing factors of heavy metals in atmospheric dust. This can provide some reference for local atmospheric environmental management. In this study, samples of atmospheric dust in Guiyang City were collected monthly (January 2012 to December 2012) and their chemical characteristics were analyzed. The concentrations of six major elements (Na, Mg, Al, K, Ca, Fe) and nine trace elements (Sc, Cr, Cu, Zn, Sr, La, Yb, Pb) in the atmospheric dust were selected to reveal the status of atmospheric pollution and to trace its natural and anthropogenic sources.

## 2. Materials and Methods 

### 2.1. Study Area

Guiyang is the central city in the Karst area of Guizhou province, southwest China (Figure 1). Its climate belongs to Subtropical monsoon humid climate with a population of about 5 million. Guiyang City has an average altitude of 1100 m and annual precipitation of 1130 mm. Because of its unique geographical location and good surrounding environment, Guiyang’s atmospheric environment is superior to most of the regional central cities in China [30].

### 2.2. Sampling Methods

The sampling site was selected as a typical urban site on the roof of a three story office building with a height of 10 m above ground near the Nanming river, Guiyang city (26°34′16″ N, 106°43′27″ E, Figure 1). Two parallel samples were collected in each sampling period using wet method procedures; the collectors were made of polypropylene [31] with a size of Φ30 cm × h40 cm. The collectors were fixed to steel shelves, mounted on 3 m tall field platforms. To depress algal and microbe growth, a collection medium consisting of 700 mL of purified glycol solution (20%) was added to each collector.

The collectors were checked routinely. Glycol solution was added as necessary to keep the bottom of the sampling containers submerged. The collectors were kept covered during rainy periods, and were exposed at all other times. Dust samples were gathered monthly. First, the solid/liquid mixture in the collector was emptied into a clean glass service; ultra-clean water and a small brush were used to remove any dust particles adhering to the collector wall. The samples were then transported to the lab for subsequent processing. After removing impurities (such as leaves and insects) by a clean plastic clip, the mixture was dried in an evaporating dish at 80 °C. 

In order to better compare and explain the source of atmospheric dust, five sampling sites around Guiyang City were selected, and 0–5 cm of topsoil samples were collected.

### 2.3. Analytical Methods

The handled dust samples were ground into powder in an agate mortar and sieved (mesh size, 75 μm), the remnants were also thoroughly ground. Then powders were dried in an oven at 105 °C for 3 h. Ultra-pure HNO_3_ and HF were used for the digestion of the dust samples, specific digestion steps are as follows:
(1)The accurate weighing 100 mg of powder samples were put into the 7 mL of Teflon digestion jar (Savillex, Eden Prairie, MN, US), then 1 mL HF and 2 mL HNO_3_ were added to the jar for digestion treatment at 140 °C for 7 days.(2)The same procedure of step 1 was reused to further dissolve the sample until the solution is thoroughly clarified. (3)After the samples are completely digested, add 2 mL HNO_3_ (1:1) two times to break up the fluorine compounds; then, dry and vaporize the samples on a hot plate. (4)Finally, dissolve the digested remainder in a 100-mL volumetric flask, using 3% HNO_3_. The digestion processes were performed in the Ultra-Clean Lab of the Institute of Geochemistry, Chinese Academy of Sciences.

## 3. Results

The major elements (Na, Mg, Al, K, Ca, Fe) concentrations of the digested solutions were analyzed using an ICP-OES (Agilent WASST-MPX, Palo Alto, California, US), the detection limit of ICP-OES is 0.01 mg L^−1^. The trace elements (Sc, Cr, Cu, Zn, Sr, La, Yb, Pb) were analyzed by an ICP-MS (Agilent 7700, Yokogawa, Japan), the detection limit of ICP-MS is 1 ng L^−1^. Both the tests were performed at the State Key Laboratory of environment geochemistry, Chinese academy of sciences. The analytical precision was estimated to be better than 5% for major elements from replicate analysis, and the precision was less than 3% for trace elements tested by multiple measurements. The content of elements in the ethylene glycol was tested, which were used as the reagent blank. The blank value was calculated and deducted according to the amount of ethylene glycol added during sampling and the amount of dust collected at last.

### 3.1. Element Content Characteristics of Atmospheric Dust

The contents of the major and trace elements (mean, ranges, standard deviation) in the atmospheric dust and of Guiyang city are listed in Table 1. Major and trace elements concentration of the dust and topsoil were different, which can be ranked by abundance as follows:
Dust: Fe > Al > Ca > Mg > K > Na > Zn > Pb > Cr > Sr > Cu > La > Sc > YbTopsoil: Fe > Al > Ca > K > Mg > Na > Zn > Cr > Pb > Cu > Sr > La > Sc > Yb

The concentrations of major and trace elements of most dusts are much higher than those of in the topsoil except for Sc and Yb, the average content of Ca and Pb in the dust reached 16.5 times and 9.92 times of the topsoil, respectively. 

In order to discriminate the groups of the elements as tracers of natural or anthropogenic sources, elements concentration data of the dust were analyzed by hierarchical cluster method. The elements were divided into three groups according to the cluster analysis (Figure 2): Ca, individually as a group, Al and Fe constitute a group, while all the rest of the elements belong to another group. This cluster pattern results is similar to the distribution results of urban soil elements in Beijing [32]. Cement is widely used in buildings and roads in urban areas. It is the main source of calcium in dust. Therefore, Ca in dust has unique geochemical characteristics. In group 2, Fe and Al were common natural elements, and their contents in various natural materials are relatively high and stable except for certain mining or smelting areas. The third group of elements was more complex, including some heavy metals which are easy to cause pollution. The content characteristics of these elements in dusts suggesting that dust is a complex mixture of natural and anthropogenic inputs, which leads to a special combination of elements in atmospheric dust. There is a significant positive correlation between calcium and strontium (*r*^2^ = 0.99, *n* = 12), which reveals that the local geological background controls atmospheric dust, although human activity affected the distribution of Ca in various particulate materials, Ca ultimately comes from the widespread carbonate rocks of this region, which is different from pure anthropogenic emissions. In addition, Ca is significant positive correlated with Mg, Sc, La and Yb (0.88 > *r*^2^ > 0.72, *n* = 12), these elements are mainly of natural sources, which further confirms that Ca in atmospheric dust was mainly from natural sources.

### 3.2. Seasonal Variation of Atmospheric Dust

There are seasonal variations of the major and trace elements concentrations in the atmospheric dust of Guiyang, the main feature is that the contents of the major and trace elements in the fall and winter are higher than those in the spring and summer. Probably because in spring and summer, good vegetation coverage and abundant rainfall make the air relatively clean. In this case, the high-density particulate (mineral matter) in the atmosphere decreases, while low-density particulate (organic matter) increases. The presence of large amounts of organic matter “dilute” the content of the major and trace elements in the atmospheric dust.

High levels of lead in dust and aerosol are generally considered to be closely related to human activity. In Guiyang, Pb concentrations of atmospheric dust in the autumn and winter are much higher than those in the spring and summer. The comparison of Pb and Ca may provide some useful evidence (Figure 3), which support the contention that some of Guiyang’s atmospheric dust is a mixture of multiple sources. The high Pb/Al ratio may be an anthropological origin source related to vehicle exhausts and coal burning, the high Ca/Al ratio may be related to the concrete buildings in the urban area, while the low Pb/Al and Ca/Al ratios are similar to those of the topsoil, which is more likely to be a natural source. The dusts in different seasons show different mixing characteristics. In spring and summer, frequent rain makes it difficult for the particulate matter of human activities to accumulate in the atmosphere and then form dust. Meanwhile, coal burning in the city and the surrounding areas decrease, thus the dust shows natural characteristics, which is similar to the surface soil. In autumn and winter, the precipitation decreases and the pollutants in air dust accumulate more easily. At this time, the coal consumption increases. These factors lead to an increase in the proportion of anthropogenic sources in autumn and Winter Atmospheric dust, and Pb/Al and Ca/Al ratio are relatively high.

### 3.3. Assessment of Atmospheric Dust Pollution

Enrichment factor (EF) was the concentration ratios of target element in the samples normalized to corresponding back ground, which was used for estimating the contribution rate of the natural and the anthropogenic sources to the samples. In this study, Al is selected as the baseline element, and the EF values of Guiyang atmospheric dust relative to the corresponding local topsoil were calculated by the formula as follows: EF_d_ = (C_ed_/CAl_d_)/(C_es_/CAl_s_). Where C_ed_ and CAl_d_ are concentrations of the element *x* and Al in dusts and C_es_ and CAl_s_ are that in the local topsoil. If the EF values close to 1, local sources are predominant and generally the values >5 indicates that non-local or anthropogenic sources were considerable. 

Guiyang belongs to the monsoon area, and the influence of wind direction on dust sources is worth considering. In winter, the northeasterly wind prevails in Guiyang. At this time topsoil-2 is in the upper wind direction of sampling point, while in summer, the southwesterly wind prevails in Guiyang, and at this time topsoil-5 is in the upper wind direction of sampling point. The EF values of each element in the dust are calculated by using the surface soil with different wind directions as background values. Although topsoil concentration values at different sites differed by a factor of 2–3 due to local effects, the same seasonality pattern of EF is obtained by using any of the five topsoil background values or their average (Figure 4). This indicates that the influence of the change of wind direction on the content of each element in the dust is not obvious. The dust in Guiyang may come more from local sources than from the long-distance transportation of wind power.

The EF values of the same element calculated from different soil backgrounds are different especially in winter, and the EF values of most elements are also the highest in one year. Northeast wind prevails in Guiyang in winter, and topsoil-2 is located in the upwind direction. The EF values of winter dust calculated with topsoil-2 as background did not show obvious anomalies, but the EF values of some elements differed by two or three times, mainly because of the heterogeneity of element contents in different topsoil. 

Considering the complexity of calculating and discussing EF values using each topsoil as the background, we used the average values of five soils as the background, and calculated EF values of each element for analysis (Figure 5). The major elements of the dust show the characteristics of enrichment relative to the topsoil, the abundances of Ca and Mg are relatively higher than the rest of the elements of dust, for their abundance values are 1.10–58.98 and 2.45–20.79, although the content of Ca and Mg in the bedrock of Karst is very high, the content of Ca and Mg in the corresponding soil is very low due to strong weathering and leaching processes. Fe and Al in the dust did not show enrichment characteristics relative to the soil. Relatively high normalized Ca and Mg values reveal that the contribution of the topsoil to the atmospheric dust was not significant.

Cu, Zn, Sr and Pb have higher abundances than the rest trace elements. The high abundance of Sr is the inheritance of its high correlation with calcium. Cu, Zn and Pb are common pollution elements which are closely related to human activities. Their high abundance in atmospheric dust indicates that atmospheric dust is likely to be partly caused by human activities; Pb has the highest abundance value of 2.35–24.86, which is likely to come from the burning process of fossil fuels in urban areas, such as boilers, automobile exhaust, and so on.

The elements that have EF values greater than 5 in the dust fall include Mg (1.19–9.81), Ca (0.53–35.72), Sr (0.69–16.37) and Pb (1.16–9.00). All the EF values greater than 5 were found in the atmospheric dust of autumn and winter, meanwhile, in those atmospheric dusts whose EF value are less than 5, showed the dust EF values in the autumn and winter were higher than those of spring and summer. Therefore, the atmospheric dust in autumn and winter is more likely to be polluted by human activity.

It is worth noting that besides the high EF value of lead showing pollution characteristics, the EF values of other conventional heavy metals were Cr (0.51–1.71), Cu (1.07–3.57) and Zn (0.45–2.33), which have not been polluted. This may be related to the lack of industry in Guiyang (Guiyang’s industrial enterprises are less than other central cities), but Guiyang also has a large number of motor vehicles. The vehicle exhaust emission is an important source of Pb pollution, which is likely to be the cause of the high EF value of Pb in Guiyang’s atmospheric dust.

The comparison of the EF values of La and Yb for atmospheric dust in Guiyang and some other areas [33,34] is shown in Figure 6. Among them, Beijing is the capital of China with heavy air pollution; Libo is a remote Kast forested area in the same province as Guiyang. It can be seen from Figure 3 that the atmospheric dusts in Guiyang and Libo are in the same interval, and have the same linear correlation, whereas the distribution and linear correlation of atmospheric dust in Guiyang and Beijing were obviously different. Suggesting that the homology of atmospheric dust in Guiyang and Libo which controlled by the carbonate geological background, thus the dust of Guiyang may mainly from natural sources.

## 4. Conclusions

The major and trace elements in the atmospheric dust of Guiyang are mostly higher than the corresponding topsoil, according to the relationship between the various elements; these elements can be divided into three groups. The behavior of Ca as a separate group may be closely related to the concrete buildings of the city; another group of elements including Fe and Al is more likely the characteristics of representative natural sources; and all of the trace elements can indicate the joint effects of natural and anthropological forces.

The element content of atmospheric dust in Guiyang showed a seasonal change in the autumn and winter higher than those in the spring and summer, this seasonal variation may be related to vegetation coverage, precipitation, and coal burning strategies in different seasons. The abundances and EF values of the falling dust relative to the topsoil revealed that only some autumn and winter atmospheric dust in Guiyang has been polluted by Pb, which may be related to a large number of city vehicle exhaust emissions. In terms of geochemical characteristics of trace elements such as La and Yb in the atmospheric dust, Guiyang is similar to the Karst forest area of Libo, which further indicates that Guiyang’s atmospheric dust mainly comes from natural processes, and a small part is affected by human activities.

## Figures and Tables

**Figure 1 ijerph-16-00325-f001:**
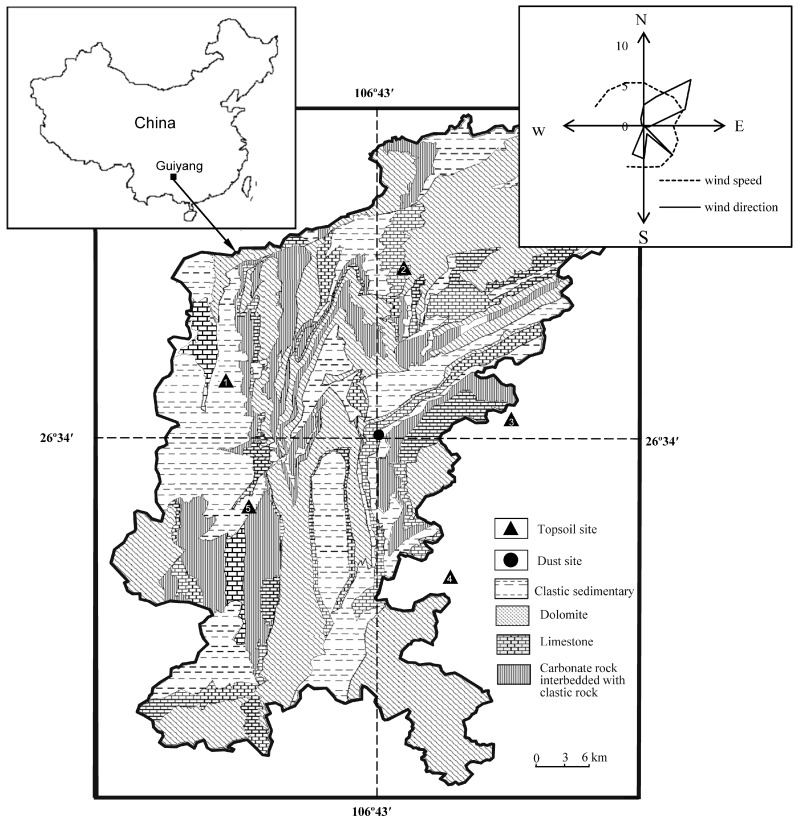
Sketch map showing the lithology of Guiyang city and sampling site.

**Figure 2 ijerph-16-00325-f002:**
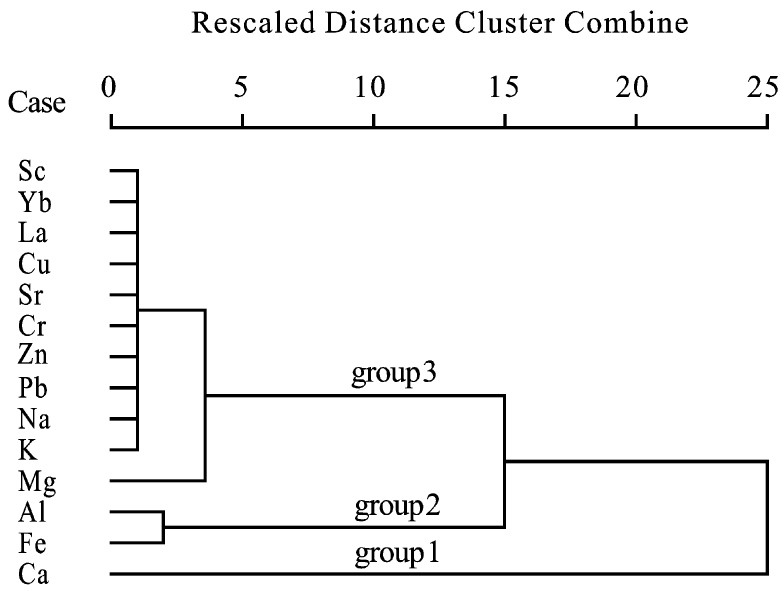
The hierarchical cluster (dendrogram) of the major and trace element concentrations in atmospheric dust of Guiyang, (*n* = 12).

**Figure 3 ijerph-16-00325-f003:**
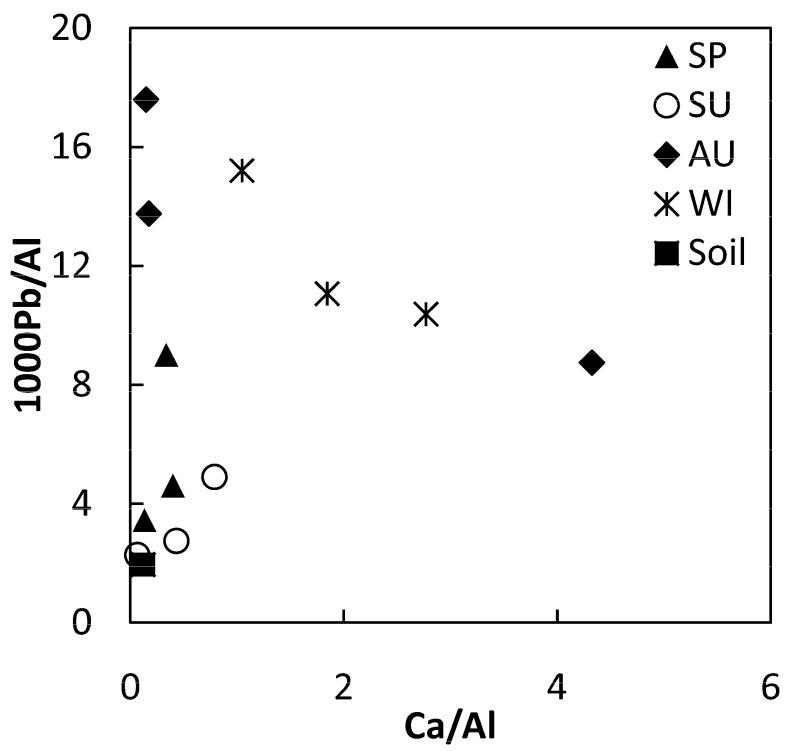
Bivariate plots of Ca/Al ratios versus 1000Pb/Al values of atmospheric dusts and averaged local topsoil of Guiyang.

**Figure 4 ijerph-16-00325-f004:**
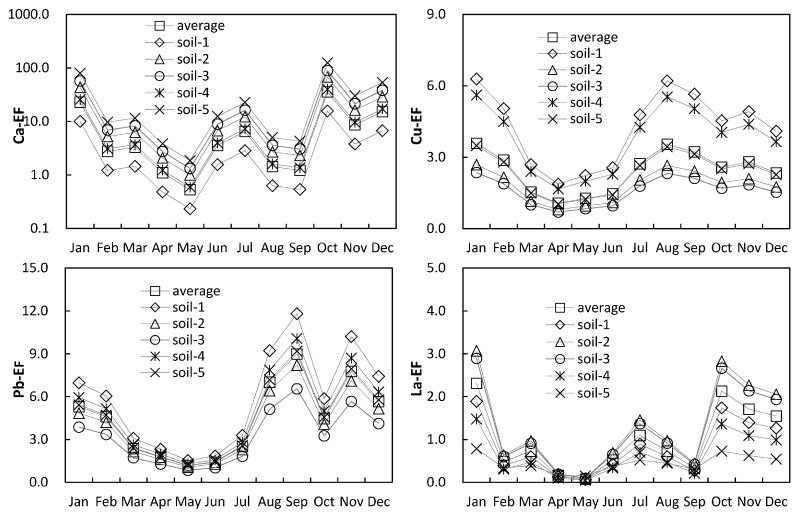
Comparison of enrichment factor (EF) values by using different topsoil background.

**Figure 5 ijerph-16-00325-f005:**
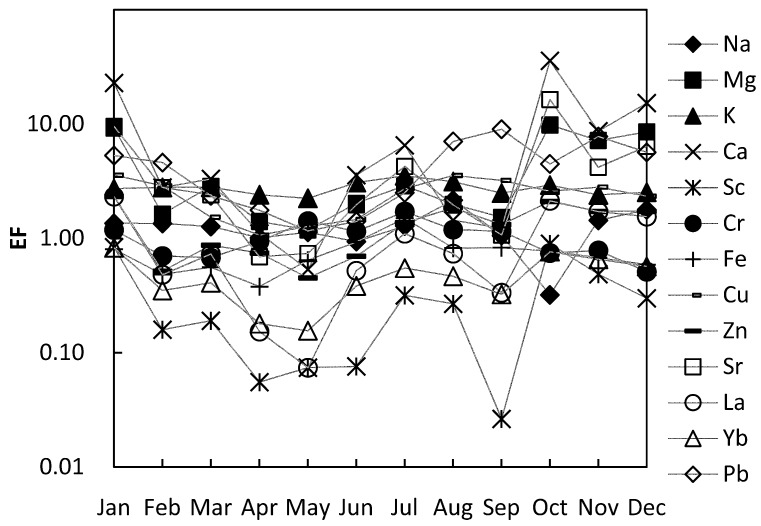
Seasonal variations of EF values of major and trace elements in atmospheric dust of Guiyang.

**Figure 6 ijerph-16-00325-f006:**
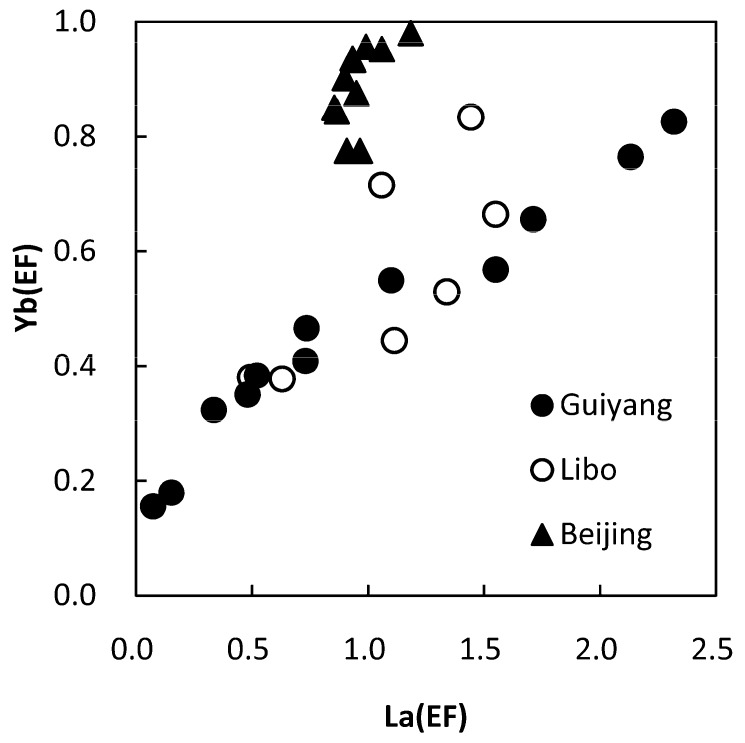
Comparison of EF relationship between La and Yb in atmospheric dust of Guiyang, Beijing [33] and Libo [34].

**Table 1 ijerph-16-00325-t001:** Concentrations of major elements (Na, Mg, Al, K, Ca, Fe. g kg^−1^) and trace elements (Sc, Cr, Cu, Zn, Sr, La, Yb, Pb. mg kg^−1^) in atmospheric dust of Guiyang.

Month	Na	Mg	Al	K	Ca	Fe	Sc	Cr	Cu	Zn	Sr	La	Yb	Pb
Jan	1.21	29.7	41.9	10.3	116	49.9	6.82	313	115	808	276	35.6	2.60	435
Feb	1.93	8.38	66.8	16.9	22.5	47.9	2.07	296	147	285	129	11.8	1.76	602
Mar	1.41	11.4	51.9	13.2	20.8	43.0	1.92	222	60.9	374	86.4	13.9	1.59	239
Apr	1.49	7.45	69.0	15.0	9.21	38.7	0.74	409	56.3	424	33.0	3.87	0.93	238
May	1.19	4.50	49.3	10.0	3.18	47.0	0.71	441	48.1	184	25.2	1.34	0.58	112
Jun	0.80	6.22	40.0	11.1	17.3	54.9	0.59	290	44.7	230	54.5	7.63	1.15	110
Jul	0.86	7.01	31.6	10.1	24.9	70.2	1.95	342	65.9	358	93.3	12.7	1.30	155
Aug	2.04	6.74	44.3	12.6	7.71	53.9	2.30	333	120	676	60.1	11.9	1.55	609
Sep	1.50	7.77	66.1	14.9	9.81	81.2	0.34	481	163	721	49.0	8.11	1.61	1163
Oct	0.27	29.8	39.5	10.5	171	45.3	6.89	186	78.1	688	448	30.9	2.27	345
Nov	1.86	33.5	60.6	13.2	63.4	60.6	5.73	300	130	865	176	38.0	2.99	922
Dec	2.21	38.2	58.2	13.4	108	50.8	3.39	188	104	831	258	33.1	2.49	644
Soil *	0.51	1.84	23.9	2.17	2.90	35.4	4.66	151	18.3	198	16.6	8.78	1.80	46.8

Soil * The mean value of five topsoil.

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
