# Peer review of "Seasonal Variation and Quality Assessment of the Major and Trace Elements of Atmospheric Dust in a Typical Karst City, Southwest China"

_ijerph, 2019, doi:10.3390/ijerph16030325_

Round 1
Reviewer 1 Report
The paper presents the results of an interesting and valuable measurement project. The methods are clearly presented, results are convincing and support the conclusions. (I cannot judge the chemical analysis methodology.)
The English grammar of the manuscript needs serious improvement, especially in the Introduction.
While this research has been performed in a high quality, one important aspect is missing: the wind. A monsoon climate is clearly dominated by two distinct wind directions (as presented in Fig. 1.). The seasonality of dust composition can therefore be caused by different source areas (besides the other factors mentioned: precipitation, coal burning etc.).
I would recommend a comparison of concentrations in seasons with predominantly northeastern and southern winds. Can a significant difference in chemical composition be shown under different wind directions? Is there a plausible explanation for the seasonality based on the wind direction (e.g. differences in land use, locations of antropogenic sources)?
Related to this: I would also recommend to present a map showing the locations of the five background sampling sites. Instead of averaging the five topsoil values, you might consider using the predominantly upwind (northeastern) sample as background. Please present the difference between topsoil concentrations at different parts around the city. Is there a significant difference between predominantly upwind and downwind soil samples, and can this be corresponded to the presented enrichment factors of each element?
Minor comments:
line 59: square meters?
line 61: please clarify this citation
Please correct the column names in table 1.
Author Response
Responses to viewer 1
Viewer 1 : While this research has been performed in a high quality, one important aspect is missing: the wind. A monsoon climate is clearly dominated by two distinct wind directions (as presented in Fig. 1.). The seasonality of dust composition can therefore be caused by different source areas (besides the other factors mentioned: precipitation, coal burning etc.).
I would recommend a comparison of concentrations in seasons with predominantly northeastern and southern winds. Can a significant difference in chemical composition be shown under different wind directions? Is there a plausible explanation for the seasonality based on the wind direction (e.g. differences in land use, locations of anthropogenic sources)?
Related to this: I would also recommend to present a map showing the locations of the five background sampling sites. Instead of averaging the five topsoil values, you might consider using the predominantly upwind (northeastern) sample as background. Please present the difference between topsoil concentrations at different parts around the city. Is there a significant difference between predominantly upwind and downwind soil samples, and can this be corresponded to the presented enrichment factors of each element?
Response: Thanks the reviewer for the helpful suggestions on our article. The impact of different wind directions and element contents of different topsoil on dust fall was mentioned by the reviewer. We also took these factors into account when we started to analyze the data and made corresponding charts for discuss, but no interesting results were obtained. Although the element content of topsoil are different, we use every topsoil as background value to calculate the EF value of each element in dust, but the seasonal variation of EF value of each element is consistent. Therefore, it is not clear that the source of dust fall can be distinguished by the difference of element content in surface soil and the change of wind direction. In the following three maps, we have made seasonal variation maps of EF values of Pb, La and Ca using every topsoil one by one as background, and obtained the same rule. Therefore, in order to simplify the calculation and discussion, the average value of topsoil are used as the background value.
Minor comments:
Viewer 1: line 59: square meters?
Response: It’s an error, ”area” should be” altitude”, we have revised it.
Viewer 1: line 61: please clarify this citation
Response: We have added the source of the citation
Viewer 1: Please correct the column names in table 1.
Response: The error has been revised.
Reviewer 2 Report
1. There are numerous grammatical errors.
2. I don’t see any mention of a field blank or control. It is important to see data from a dust collector filled with glycol and left closed in the field to show no metal content.
3. Further, there is no mention of uncertainty, including: recovery efficiency from collector, instrumental uncertainty from the ICPs used, limit of detection, etc. The samples were collected in duplicate; they should have been analyzed in duplicate – were they just mixed together? Parallel analysis would give another estimate of uncertainty.
I would only endorse publishing after these three items are addressed.
Small notes:
1. Define EF in the abstract.
2. 58-59: What is the average area of a city? Is the area 1100 square meters?
3. 77-78. Was there any kind of screen on the collector to keep out leaves or insects, or was it completely open? Do leaves and insects contribute to any of the metals measured?
4. Table 1: The headings don’t line up with the numbers in the table. Draw a dividing line between major and trace elements (Fe / Sc) because the units are different.
p { margin-bottom: 0.1in; line-height: 120%; }
Author Response
Responses to viewer 2
Viewer 2: There are numerous grammatical errors.
Response: We have corrected the grammar of the article.
Viewer 2: I don’t see any mention of a field blank or control. It is important to see data from a dust collector filled with glycol and left closed in the field to show no metal content.
Response: At the beginning of the data analysis, all the test data have deducted the ethylene glycol blank. We have added a description of the test and quality control of the blank value of ethylene glycol to this manuscript.
Viewer 2: Further, there is no mention of uncertainty, including: recovery efficiency from collector, instrumental uncertainty from the ICPs used, limit of detection, etc. The samples were collected in duplicate; they should have been analyzed in duplicate – were they just mixed together? Parallel analysis would give another estimate of uncertainty.
Response: According to the reviewer's suggestion, we describe the detection limit and test error of the instrument in the section of the method. Because of the small amount of samples collected and the fact that another parallel sample has been used for other analysis, we have not done parallel experiments.
Viewer 2: Define EF in the abstract.
Response: EF in the abstract has been adopted as its full name
Viewer 2: 58-59: What is the average area of a city? Is the area 1100 square meters?
Response: It’s an error, ”area” should be” altitude”, we have revised it.
Viewer 2: 77-78. Was there any kind of screen on the collector to keep out leaves or insects, or was it completely open? Do leaves and insects contribute to any of the metals measured?
Response: Sampling in the field is open, but in the pre-treatment of solid-liquid mixtures, we use manual methods to clean up the leaves and other debris, which has been mentioned in section of “sampling methods”.
Viewer 2: Table 1: The headings don’t line up with the numbers in the table. Draw a dividing line between major and trace elements (Fe / Sc) because the units are different.
Response: We have corrected the errors in the table and clearly labeled the major and trace elements in the title.

Round 2
Reviewer 1 Report
Thank you for the reply and the figures.
"The impact of different wind directions and element contents of different topsoil on dust fall was mentioned by the reviewer. We also took these factors into account when we started to analyze the data and made corresponding charts for discuss, but no interesting results were obtained."
If I understand well, this states that no difference was found in the dust composition among different wind directions. This IS an interesting result indeed! It's a strong sign that the air pollution is caused by local sources rather than large-scale transport (that would bring different chemical footprints from different directions). I think this is an interesting result and should be included in the paper. If you decide not to, at least a short note should be placed in the text that you investigated the impact of wind direction and it is not the reason of seasonality.
Regarding the topsoil issue: it's good to see in the figures that the seasonality is the same at all sites. However, there is a difference of a factor of 2-3 among values at different sites (that shows this range of difference in topsoil concentration values). The difference is probably caused by local emissions. It is showed by the fact that the order of lower-to-upper concentrations are different for different elements, e.g., Soil 3 seems Pb-rich while Soil 5 is La-rich. A "clear" upwind and "polluted" downwind pattern should show the same order of concentrations among stations for all elements. This raises questions on how "background" these stations actually are, as normally background sites should not be prone to local effects. While I accept yours as the best possible solution as finding a true background site near a megacity is impossible, I recommend to add a paragraph elaborating on this problem or at least a sentence like this to clarify the topsoil issue:
"Although topsoil concentration values at different sites differed by a factor of 2-3 due to local effects, the same seasonality pattern of EF was obtained by using any of the five topsoil background values or their average."
I also recommend to publish the EF seasonality charts for all elements (like the ones that you included in your reply) as an appendix of the paper, coupled with a map showing the locations of topsoil sample sites. They are interesting and meaningful!
Author Response
Responses to viewer 1
Viewer 1: "The impact of different wind directions and element contents of different topsoil on dust fall was mentioned by the reviewer. We also took these factors into account when we started to analyze the data and made corresponding charts for discuss, but no interesting results were obtained."
If I understand well, this states that no difference was found in the dust composition among different wind directions. This IS an interesting result indeed! It's a strong sign that the air pollution is caused by local sources rather than large-scale transport (that would bring different chemical footprints from different directions). I think this is an interesting result and should be included in the paper. If you decide not to, at least a short note should be placed in the text that you investigated the impact of wind direction and it is not the reason of seasonality.
Regarding the topsoil issue: it's good to see in the figures that the seasonality is the same at all sites. However, there is a difference of a factor of 2-3 among values at different sites (that shows this range of difference in topsoil concentration values). The difference is probably caused by local emissions. It is showed by the fact that the order of lower-to-upper concentrations are different for different elements, e.g., Soil 3 seems Pb-rich while Soil 5 is La-rich. A "clear" upwind and "polluted" downwind pattern should show the same order of concentrations among stations for all elements. This raises questions on how "background" these stations actually are, as normally background sites should not be prone to local effects. While I accept yours as the best possible solution as finding a true background site near a megacity is impossible, I recommend to add a paragraph elaborating on this problem or at least a sentence like this to clarify the topsoil issue:
"Although topsoil concentration values at different sites differed by a factor of 2-3 due to local effects, the same seasonality pattern of EF was obtained by using any of the five topsoil background values or their average."
I also recommend to publish the EF seasonality charts for all elements (like the ones that you included in your reply) as an appendix of the paper, coupled with a map showing the locations of topsoil sample sites. They are interesting and meaningful!
Overall this manuscript is much improved with addition of the blanks, controls, and detection limits. But there are still many grammatical errors. I would suggest a thorough editing for grammar.
Response: Thanks the viewer for the valuable analysis experience. In the discussion part of the article, the influence of wind direction and surface soil on atmospheric dust fall is analyzed, and the corresponding charts are given, and the results are discussed. The grammar has been further modified.

Reviewer 2 Report
Overall this manuscript is much improved with addition of the blanks, controls, and detection limits. But there are still many grammatical errors. I would suggest a thorough editing for grammar.
line 61, latitude should be altitude.
Author Response
Viewer 2: Overall this manuscript is much improved with addition of the blanks, controls, and detection limits. But there are still many grammatical errors. I would suggest a thorough editing for grammar.
Response: We have made further modifications to the grammar of the article.
Viewer 2: line 61, latitude should be altitude.
Response: Yes, We have revised it.
